# Multi-Layered Non-Local Bayes Model for Lung Cancer Early Diagnosis Prediction with the Internet of Medical Things

**DOI:** 10.3390/bioengineering10020138

**Published:** 2023-01-20

**Authors:** Yossra Hussain Ali, Seelammal Chinnaperumal, Raja Marappan, Sekar Kidambi Raju, Ahmed T. Sadiq, Alaa K. Farhan, Palanivel Srinivasan

**Affiliations:** 1Department of Computer Sciences, University of Technology, Baghdad 10066, Iraq; 2Department of Computer Science and Engineering, Solamalai College of Engineering, Madurai 625020, India; 3School of Computing, Sastra Deemed University, Thanjavur 613401, India

**Keywords:** Internet of Things, medical things, deep learning, naive Bayes, machine learning, diagnosis prediction, lung cancer

## Abstract

The Internet of Things (IoT) has been influential in predicting major diseases in current practice. The deep learning (DL) technique is vital in monitoring and controlling the functioning of the healthcare system and ensuring an effective decision-making process. In this study, we aimed to develop a framework implementing the IoT and DL to identify lung cancer. The accurate and efficient prediction of disease is a challenging task. The proposed model deploys a DL process with a multi-layered non-local Bayes (NL Bayes) model to manage the process of early diagnosis. The Internet of Medical Things (IoMT) could be useful in determining factors that could enable the effective sorting of quality values through the use of sensors and image processing techniques. We studied the proposed model by analyzing its results with regard to specific attributes such as accuracy, quality, and system process efficiency. In this study, we aimed to overcome problems in the existing process through the practical results of a computational comparison process. The proposed model provided a low error rate (2%, 5%) and an increase in the number of instance values. The experimental results led us to conclude that the proposed model can make predictions based on images with high sensitivity and better precision values compared to other specific results. The proposed model achieved the expected accuracy (81%, 95%), the expected specificity (80%, 98%), and the expected sensitivity (80%, 99%). This model is adequate for real-time health monitoring systems in the prediction of lung cancer and can enable effective decision-making with the use of DL techniques.

## 1. Introduction

The use of sophisticated technologies has modified traditional healthcare practices, with practical and high-quality results. Such technologies provide effective systems for predicting lung cancer, and have been developed in massive numbers in recognition of the value of their distinct outcomes [1]. The deployment of an intelligent healthcare system can provide effective results that are comparable to those of the existing systems in terms of their efficiency and accuracy in regard to their end-goals. The emergence of these new technologies has promoted a system with sensitivity in the image detection process, leading to high-quality results. In healthcare practice, artificial intelligence (AI) systems can be deployed to provide functional support and their results can be compared with those obtained using a traditional approach. Such an approach provides an effective decision-making process, with precise results developed through AI [2]. One image processing technique, the DL process, has been found to provide effective and high-quality results [3]. A research process using a conceptualized framework for the early prediction of disease would be supportive in a medical setting. Such prior predictions could lead clinicians to take further steps to diagnose lung cancer in cases without any additional indications. Effective results can be obtained with modern technologies such as the IoMT approach with DL and the NL Bayes model [4]. Such an approach could be effective compared to the traditional approach, and it may be possible to avoid developmental delays in predicting lung cancer through the development of an associated framework.

Some practical medical processing steps are involved in the prediction and diagnosis of the evolution of lung cancer, such as:i.Processing images—removing unwanted data or noise that are present in the image [5].ii.Extracting images—converting images into specific groups or sectors of input resources.iii.Choosing features—enhancing the response time and execution time compared with the distinct resultant values of image processing.

The DL Mask R-CNN model was developed to predict lung cancer with categorization and classification using image segmentation applied to pulmonary nodules [6]. Lung cancer prediction is effective in lung cancer diagnosis when distinct factors are used and resultant quality values are reached. It is important to use effective and quality factors and outsourcing, in order to distinguish between different results. The DL network represents as an effective system for categorizing functions for image recognition. This technique could enhance the performance of the prediction method, providing excellent results. The detection of the image could be carried out with the aim of determining the disease at an early stage [7]. The image delivered at the end-stage of this process would be a high-resolution image with spatial information and modalities. Convolutional neural networks (CNNs) can be used to categorize benign and malignant tissues through the use of CT scan images. The lung cancer prediction process would benefit from the use of a high-performance technique with a reduced cost and the ability to be widely propagated in order to provide excellent results. The process of medical imaging was developed to diagnose lung cancer through the use of CT scan images. The X-ray, CT, and MRI processes are supporting imaging technologies, used to predict disease occurrence, and their use has evolved as part of the medical process. For this reason, imaging [8,9,10], machine learning (ML) [11], and biosensor technology [12] methods were used in this study to identify lung cancer.

## 2. Literature Survey

Lung cancer can be predicted at the critical phase where it cannot be diagnosed. The necessary steps must be considered as part of the prediction process before a determination can be made. Over time, an approach to predicting the presence of lung cancer through image processing and the extraction of features from images has evolved [13,14,15,16]. Lung cancer can be predicted by applying AI techniques [17]. However, unsatisfactory results were obtained due to the system failing to categorize domain and system function events in the different results. This approach failed to provide information on lung cancer in comparison with the traditional method with basic technological features in relation to distinct factors [18,19]. Various functional values and solutions to manage, contribute, and define factors for the detection process have been developed [20,21,22]. The CT process has been used to predict the diagnostic factors of lung cancer, with promising results [23,24,25]. False factors occurring in distinct sectors of CT scan images have also been observed in the determination of attributes. The detection of disease factors has been carried out through the CAD of lung cancer. The use of this system is based on obtaining reliable information on the activity of distinct factors to obtain high-quality results.

The DL process employs an effective source to demonstrate the expected result in the prediction of lung cancer [26]. It provides specific, efficient results to obtain accurate results. The purpose of reviewing different models is to categorize different events and functional aspects. Nodule detection and false-positive reduction systems have been modeled using DL algorithms to detect lung cancer [27]. A neural network model was trained on some whole-slide images, using DL for lung cancer classification [10,28]. This area of research aims to define the quality and early prediction capabilities of systems using slide training with pathological approaches. The system has been evolving, with the use of scaling factors and training system activities to obtain the desired results. It is necessary to use a defined training program with a neural network to develop an action plan. The necessity to determine this action plan could be supportive for detecting the value of early signs of disease according to their types. This approach could be used to sort clarified result values according to their efficiency and their function and to avoid human losses.

ML techniques have been applied to identify lung cancer in high-risk individuals [11,29]. In this system, a model differentiating between benign and malignant tissues is used to differentiate layers and predict disease more easily. This system focuses on deducing computational cost factors with efficient result values to emphasize the distinct functional values accomplished. The functional factors of the network are densely interconnected to maintain quality and ensure that efficient results are obtained. This approach aims to improve neuron value and drops out particular layers. The performance values of CT scans could be optimized by managing the approach of developing distributive functional terms and evolved values.

The CAD model was developed to enhance and improve the operational value and performance of the system. The objectives of this approach are to provide an accurate diagnosis and treatment process for a medical imaging system. Medical images from a CT scan are used to determine the occurrence of disease in the lungs. In the CT scan, a source factor is implemented with a DL approach to obtain on high-quality results. The tools and techniques that have evolved to manage these activities and medical analyses are effective in accomplishing defined values. The CNN model was developed to provide an automatic and adaptive perspective, deploying the methodology of a directed approach to obtain the desired results. This model effectively sorts quality result values when implemented with a computer vision process developed according to radiology factors. This approach could be effective and efficient in improving the quality of the resulting values and the functioning of the system according to the evolved radiology and application approach. It acts as a building block, with multiple adequate resources used to resist defined values. The use of a neural network is essential to enhancing the potential activity carried out to resist the determined deliverable [29,30].

The primary knowledge factor was introduced with the image processing model to predict disease factors through an image processing system. This approach supports the prediction by means of the ML process, which can be categorized into two approaches—the supervised ML and the unsupervised ML process. The supervised ML process requires a sustained manual intervention to obtain accurate results [8,31]. The unsupervised ML process does not require any manual intervention to proceed with the operational process. These factors evolved along with the influential factor of the system function, resulting in an appropriate decision-making process and retaining a high-quality system by managing evident factors to reach distinct result values. It consists of a practical process of determining and segmenting images for diagnosis according to their layers [9,32,33]. The sectoring layers of CT scans are necessary to predict lung disease using CT scan images. CNNs and DL act as effective sources to predict desired values and provide accurate results in the decision-making process. The use of an AI system integrated with IoMT to enhance their performance through sustainability and scalable factors has been proposed here in relation to lung cancer prediction.

## 3. Proposed Model

The classification of the disease factor prediction process is illustrated in Figure 1. A functional diagram of the lung cancer prediction process is depicted in Figure 2. The prediction classification process is modeled in Figure 3.

The proposed model predicts lung cancer through the use of IoMT with a DL system. This methodology involves categorizing the events into sectors for propagating resultant quality values effectively when categorizing concern events and processes according to their domain to develop an accurate result. A block diagram of the proposed model is shown in Figure 4. The functional architecture of the proposed model includes input resources, the preprocessing sector, an extraction and classification domain, sensor devices, and classification.

### 3.1. Preprocessing

A set of input values is developed in the initial phase of this model. Input resources (CT scan images) are collected. The system is designed to manage and develop a distinct resultant value through the image processing system. Before proceeding with the image processing aspect, it is necessary to remove unwanted noise or distortion that occurs in the input. This process penetrates the system’s value with a resultant practical value to accomplish a high-quality value, which is used to demonstrate the quality result. In this process, deep detection is carried out with a defined sector to accomplish and determine the quality outsource value. This can provide an effective and efficient source for image processing and the categorization of distinct resultant values to avoid confusion in predicting cancer in the lungs through the use of CT scan images.

### 3.2. Feature Extraction

The extraction approach evolves and categorizes CT scans. It acts as a source for the processing system with its distinguishing factor being the process implemented to manage CT scans of the lungs. This approach plays a vital role in a propagating system with a defined structure to stimulate the attributes of valid and invalid factors through the detection process, which is capable of the distinct development of resource values and functions through their risk state.

### 3.3. Domain Classification

The classification of CT scan images is based on the occurrence of disease factors that are designed to retain appropriate decision-making values. The classification of the system design model is depicted in Figure 5. The decision-making is carried out by utilizing a DL process with effective and quality results and outcomes. A dedicated matrix has been developed to manage the deciding factors according to the prediction function through the naive Bayes system model, which effectively determines the sensitive value and defined functional statements. It proliferates the value-of-concern approach, which is expressed to obtain early-stage predictions of cancer occurrence in the lungs with appropriate error rate values. It can be managed and compelled to state practical values. On the basis of this consideration, the resulting value can support effective decision-making through the determination of distinct values, and it is efficient in the prediction process to retain the quality outsource factor. The multi-layer image prediction model plays a vital role in demonstrating the key factors used to recognize defined events and functional values for the image prediction system. It penetrates the defined value when considering the eventual function of predicting disease factors through determined methods.

### 3.4. Algorithm of Proposed Model

NL-Bayes is a sophisticated form of NL. Every patch in the NL method is replaced with the weighted means of the community’s maximum comparable patches. Because images are often self-similar, times of comparable patches are frequently detected, and averaging them increases the SNR. The NL-Bayes method improves on the NL-way method by assessing a Gaussian vector model for each group of related patches. As a consequence, each patch has a mean, as well as a covariance matrix that estimates the patch organization’s variability. The Gaussian patch means are implemented in equal iterations, but the second iteration uses the denoised photos from the first generation to estimate the implied higher covariance. A flowchart of the proposed model is presented in Figure 6.
**Algorithm 1:** Proposed Model1: Let $a_{0}$ be an ideal image. Define the noisy image corrupted with Gaussian noise ‘*a*’, noise factor *n*, and standard deviation *σ* using$a=a_{0}+n$(1)2: Evaluate the conditional probability distribution using$P\left(a_{0}/a\right)={\left(2\pi \sigma^{2}\right)}^{-1}e^{-\left({\left(a-a_{0}\right)}^{2}/2\sigma^{2}\right)}$(2)3: Determine the observed noise version P*_n_* given P_0_, the noiseless patch of *a*_0_ using$P\left(P_{n}/P_{0}\right)=ce^{-\left({\left(P_{n}-P_{0}\right)}^{2}/2\sigma^{2}\right)}\;$where $c={\left(2\pi \sigma^{2}\right)}^{-1}$(3)4: Define the Euclidean norm of $P_{0}$ as $\left\|P_{0}\right\|$. Compute $P\left(P_{n}/P_{0}\right)$ using the Bayes rule.$P\left(P_{n}/P_{0}\right)=\;P\left(P_{n}/\|P_{0}\|\right)P\left(\left\|P_{0}\right\|\right)/P\left(P_{n}\right)$(4)5: For a normalization constant α, the initial cluster $Q_{0}$, iteration *t*, find the direct path, P(Q) to construct cluster Q of Gaussian samples.$P\left(Q\right)=\alpha cP_{0}e^{\left(Q_{0}-P_{0}\right)t-1}\left(Q_{0}-P_{0}\right)/2$(5)6: Evaluate $\arg P_{0}\max P\left(P_{n}/P_{0}\right)$.$\arg P_{0}\max P\left(P_{n}/P_{0}\right)=\arg P_{0}\max\;e^{-{\left(\|P_{0}-P_{n}\|\right)}^{2}}cP_{0}\|P_{0}-P_{n}\|{)}^{2}/4\alpha^{2}$(6)7: Find the posterior estimation $C_{p_{n}}.$$C_{p_{n}}≅C_{p_{0}}+\alpha^{2}$(7)8: Determine the maximum posterior estimation using$\arg P_{0}\max P\left(P_{n}/P_{0}\right)=\arg P_{0}\max\;e^{-{\left(\|P_{0}-P_{n}\|\right)}^{2}}cP_{0}\|P_{0}-P_{n}\|{)}^{t}/\alpha^{2}$(8)9: Evaluate $C_{p_{n}}.$$C_{p_{n}}=P_{n}+\left(C_{p_{\left(n-1\right)}}-\alpha^{2}\right)cP_{n}\left(P_{n}-P_{0}\right)$(9)10: Update $C_{p_{n}}$ using$C_{p_{n}}≅\frac{1}{P{\left({\hat{p}}_{n}\right)}^{-1}}\underset{\hat{Q}\in P\left({\hat{p}}_{n}\right)}{\sum }\left({\hat{Q}}_{\;}-\bar{{\hat{P}}_{n}}\right)\;{\left({\hat{Q}}_{\;}-\bar{{\hat{P}}_{n}}\right)}^{t}Q$(10)11: Estimate $C_{p_{0}}\;$using sampling estimates, neighbor patches ${\hat{Q}}_{1}$.$C_{p_{0}}=\frac{1}{P{\left({\hat{p}}_{1}\right)}^{\;}}\underset{\hat{Q}\in P\left({\hat{p}}_{1}\right)}{\sum }\left({\hat{Q}}_{1}-\bar{{\hat{P}}_{1}}\right)\;{\left({\hat{Q}}_{1}-\bar{{\hat{P}}_{1}}\right)}^{t}$(11)$\bar{{\hat{P}}_{1}}≅\frac{1}{P\left({\hat{p}}_{1}\right)}\underset{\hat{Q}\in P\left({\hat{p}}_{1}\right)}{\sum }{\hat{Q}}_{1}$(12)12: Approximate $C_{p_{n}}$ using the covariance matrix of noisy patches $c_{p_{n}}=c_{p_{0}}+\nabla^{2}I$(13)13: Apply the Bayes method.${\hat{P}}_{2}=\bar{{\hat{P}}_{1}}+c_{{\hat{p}}_{1}}{\left[c_{{\hat{p}}_{1}}+\nabla^{2}I\right]}^{-1}(P_{n}-{\bar{P}}_{n1})$(14)14: Perform grey level image—denoising process for different values of σ.**Type**σ**= 2**σ**= 5**σ**= 10**σ**= 20**σ**= 30**σ**= 40**σ**= 60**σ**= 80**σ**= 100**Classic46.1440.8935.5732.4630.3127.6326.0326.0325.41Iteration46.1940.1236.6233.6029.6727.5126.2426.2425.72Patch size:$e_{1}=\{\begin{array}{c} 3,\;\;\;\;\;\;\;\;\;\;\;\;\;\sigma \leq 20\\ 5,\;\;\;20<\sigma \leq 50\\ 7,\;\;\;\;\;\;\;\mathrm{otherwise} \end{array}$ $e_{2}=\{\begin{array}{c} 3,\;\;\;\;\;\;\;\;\;\;\;\;\;\sigma \leq 50\\ 5,\;\;\;50<\sigma \leq 70\\ 7,\;\;\;\;\;\;\;\mathrm{otherwise} \end{array}$(15)15: Find a similar pattern retained:$N_{1}=3e_{1}^{2}$ $N_{2}=3e_{2}^{2}$(16)16: Compute the size of the search area:$\lambda_{1}=$ $N_{1}/2$, $\lambda_{2}=N_{2}/2$(17)17: Set the threshold as$\tau_{0}=16e_{2}^{2}$(18)$C_{p_{1}}≅\frac{1}{P{\left({\hat{p}}_{n}\right)}^{-1}}\underset{\hat{Q}\in P\left({\hat{p}}_{n}\right)}{\sum }\left({\hat{Q}}_{1}-\bar{{\hat{P}}_{n}}\right)\;{\left({\hat{Q}}_{1}-\bar{{\hat{P}}_{n}}\right)}^{t}$(19)18: Estimate ${\hat{P}}_{2}$ using${\hat{P}}_{1}=\bar{{\hat{P}}_{n}}+c_{{\hat{p}}_{n}}{\left[c_{{\hat{p}}_{n}}+\sigma^{2}I\right]}^{-1}\left(P_{n}-{\bar{P}}_{n}\right)$, ${\left({\hat{P}}_{n}\right)}^{-1}≅\frac{1}{P\left({\hat{p}}_{1}\right)}\underset{\hat{Q}\in P\left({\hat{p}}_{1}\right)}{\sum }\tilde{Q}$(20)$C_{{\hat{P}}_{1}}≅\frac{1}{P{\left({\hat{p}}_{1}\right)}^{-1}}\underset{\hat{Q}\in P\left({\hat{p}}_{1}\right)}{\sum }\left({\hat{Q}}_{1}-{\left({\hat{P}}_{n}\right)}^{-1}\right){\left({\hat{Q}}_{1}-{\left({\hat{P}}_{n}\right)}^{-1}\right)}^{t}$(21)${\hat{P}}_{2}={\left({\hat{P}}_{n}\right)}^{-1}+c_{{\hat{p}}_{1}}{\left[c_{{\hat{p}}_{1}}+\sigma^{2}I\right]}^{-1}\left(P_{n}-{\left(P_{n}\right)}^{-1}\right)$(22)19: Evaluate the huge set of original patches using minimum mean square estimation P(PnP^n)=1(2πσ2)e‖P0−Pn2‖2σ2(23)P^0=E[PnP^n]=∫P(PnP^n)P0dP0=∫P(P0pn)P(pn)P(P0)P0dP0(24)P^0≅1m∑iP(PnP0)P0i1m∑iP(PnPi)(25)20: Evaluate E(P0Pn) using wavelet neighborhood de-noising.$P_{0}=\sqrt{zU}$(26)$P_{n}=P_{0}+N=\sqrt{zU}+N$(27)E(P0Pn)=∫0∞P(zPn)E(P0Pn,z)dz(28)$\;\arg\underset{\alpha }{\min}{\left\|P\right\|}_{0}$ = $\|P_{0}-D_{\alpha }\|{}_{\;}^{2}\leq e^{2{\left(c\sigma \right)}^{2}}$(29)21: Perform the restoration using$\;P\left(P_{i}\right)=\frac{1}{{\left(2\pi \right)}^{\frac{e^{2}}{2}}{\left|c_{e_{i}}\right|}^{\frac{1}{2}}}\;e^{-\frac{1}{2}{\left(P_{i}-µ\right)}^{T}c_{e_{i}}\left(P_{i}-µ\right)}$(30)22: Calculate $P_{i}$ with index $k_{i}$ using(Pi,ki)=arg maxpklog P(PiPni) (31)(Pi,ki)=arg maxpk(logP(PniPi,ck)+logP(Pick)) (32)

## 4. Simulation and Analysis

A comparison of the set of rules reveals that it is similar in spirit to numerous modern algorithms (TSID, BM3D, BM3D-SAPCA) and has a very similar structure to BM3D. The extensive experimental assessment carried out in this study revealed that the set of rules delivered the best state of the art in terms of PSNR and picture quality using color images. Most patch-based photo-denoising approaches may be summed up in a single paradigm that combines the remodel thresholding methodology with Markovian Bayesian estimation. This unification is complete when the patch space is considered a Gaussian mixture. The orthonormal basis of patch eigenvectors is related to each Gaussian distribution. On these local orthogonal bases, transform thresholding was performed. The method presented in this study maintains the most intriguing elements of previous methodologies, while marginally improving upon the quality of more excellent image methods. The proposed model was simulated for training on 80% of the dataset and for testing on 20% of the dataset, with the classifiers on the clinical dataset, which consisted of a set of 15,000 clinical images containing 6782 benign and 8218 malignant lung cancer images.

### 4.1. Preprocessing Analysis

The results have been summarized with a distinct approach to propagating the resultant quality values of the processed images. The preprocessing stage is the initial phase in the development of a model with the aim of obtaining an effective and quality result. Emphasizes the accomplishment of the functional activity involved in this phase, the system was designed to remove experimental noise factors. These stages were determined within the system according to the quality results of the filtering model and determining the quality outcomes of predicting disease in an earlier stage when compared with practical factors. It may be sufficient to reach valid information factors to reach out to a determined sector of influential information factors.
(33)y(m, n)=median (x(i, j) such that (i, j)∈τ)
(34)τ=neighbor(m, n)

This approach was used to evaluate the differentiating equation of the noise removal process accomplished, with a practical outcome of reaching quality determinants.

### 4.2. Image Level Balancing

The process of removing noise values was carried out via the normalization of the system function through determining the included equation factors, as follows:(35)$$I^{*}=a_{new}+\left(b_{new}-a_{new}\right)\;\left(I-a\right)/\left(b-a\right)$$
where $\left(b_{new}-a_{new}\right)$ defines the normalization of a new image factor.

### 4.3. Noise Removal Analysis

After the preprocessing system was completed, the defined sector of the completed outcome was penetrated to reach defined values according to the quality results. Figure 7b shows a CT scan image of noise removal, Figure 8 shows the normalization result and Figure 9 presents the finalized outcome of lung cancer prediction.

Table 1 shows the features and descriptions of information of types and system functions that evolved among distinct aspects. The accuracy of the model was defined using the elements of the confusion matrix, with true negative (TN), true positive (TP), false negative (FN), and false positive (FP) values. This measure expresses the classifier performance of valid predictions based on the entire dataset.
(36)Accuracy=(TN+TP)/(FN+FP+TP+TN)
(37)Specificity=TN/(FP+TN)
(38)Sensitivity=TP/(FN+TP)

### 4.4. Results and Discussion

The performance measures of the proposed model were compared with the results for some of the other models [1,2,3,4,5,6,7,8,34,35,36,37,38,39,40,41,42]—the stochastic gradient descent, random forest, k-nearest neighbors, decision tree, naive Bayes, multi-layer perceptron, logistic regression (LR) and support vector machine methods, as shown in Table 2, Table 3 and Table 4. The following inferences were obtained from the simulation.

▪The proposed model achieved the expected accuracy (81%, 95%), the expected specificity (80%, 98%), and the expected sensitivity (80%, 99%) for the considered datasets and provided better results than the other models [1,2,3,4,5,6,7,8,41,42].▪The best typical values of σ for the gray-level image de-noising process lay in (2, 100).▪87% of the high risk was detected with the highest sensitivity (TP rate) and specificity (TN rate) of 98% compared to the LR models.▪The proposed model provided a low error rate (2%, 5%) and an increase in the number of instance values.▪The range for smaller patch sizes was randomly defined with three intervals. When the patch size exceeded 2 with three intervals for the considered high-resolution images, variations were observed in restoration and performance measures. Hence, the patch size was set as 2, and the best random intervals were obtained through the simulation.

## 5. Conclusions and Future Work

Thus, the use of DL techniques in the detection and diagnosis of lung cancer is practical and can be differentiated from the traditional method. It acts as an effective method to process a system with distinct factors, enabling users to recognize the values of certain eventually identified factors. This approach enabled us to develop an effective source of results to manage and accommodate the values of the defined sector. The method may be effectively improved by accomplishing the distinct aspects identified here. The research process was carried out to determine the value of dependence on the quality outcomes. The approach may be used to detect the effects of disease early and thus avoid human losses. The effectiveness of defining excellent outcomes was propagated through the distinct approach using sustained quality result values to retain different systems and to sort quality values, which were propagated to reach distinct outsourcing applications. The proposed model achieves the expected accuracy (81%, 95%), the expected specificity (80%, 98%), and the expected sensitivity (80%, 99%) for the considered datasets and provided better results than the other models [1,2,3,4,5,6,7,8]. The proposed model provided a low error rate (2%, 5%) and an increase in the number of instance values.

For this reason, this technique was deployed to emphasize the outcome of early detection, with accurate results and values as propagated. Different evolutionary and recommendation models can be applied in the future to improve the expected performance measurements [33,34,35,36,37,38,39,40]. Hybrid DL models can also be developed to obtain better performance measurements in a reduced time [41,42].

## Figures and Tables

**Figure 1 bioengineering-10-00138-f001:**
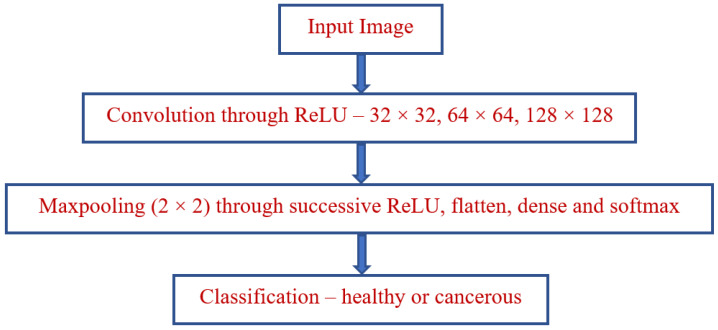
Classification—disease factor prediction.

**Figure 2 bioengineering-10-00138-f002:**
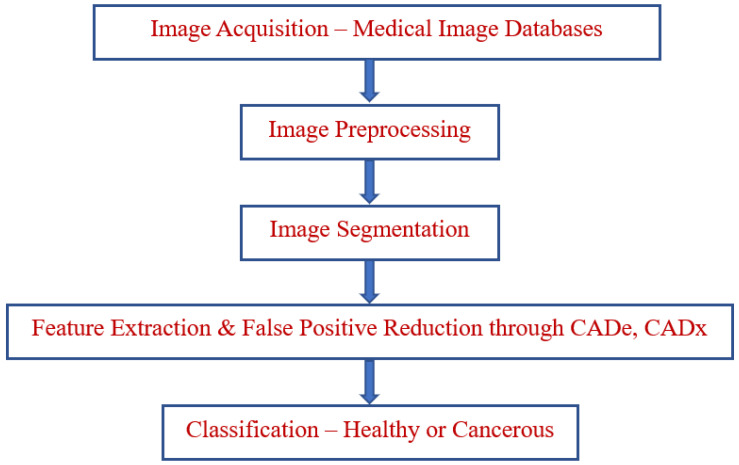
Lung cancer prediction—functional diagram.

**Figure 3 bioengineering-10-00138-f003:**
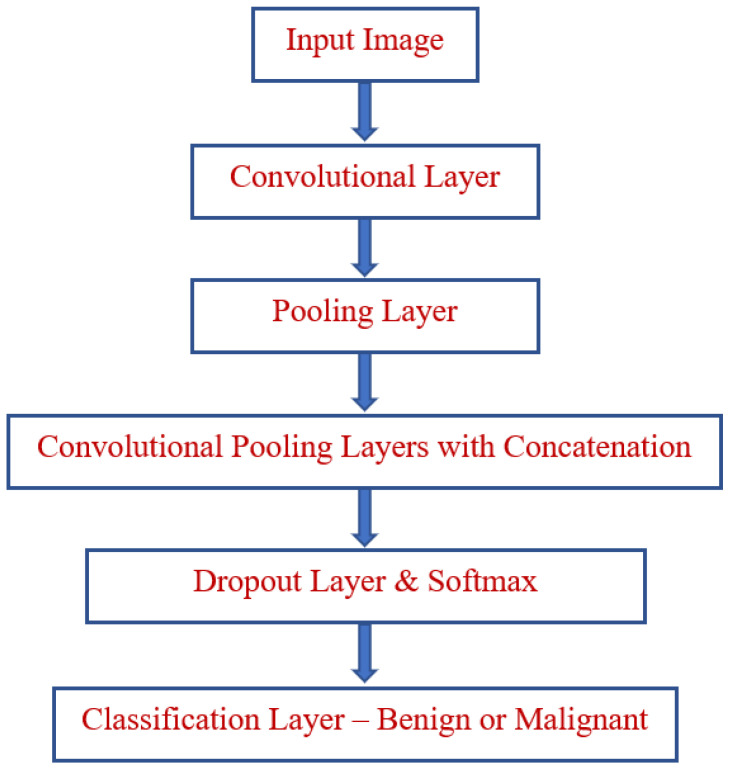
Prediction classification.

**Figure 4 bioengineering-10-00138-f004:**
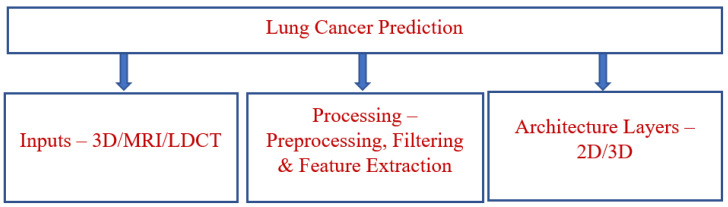
Block diagram of DL.

**Figure 5 bioengineering-10-00138-f005:**
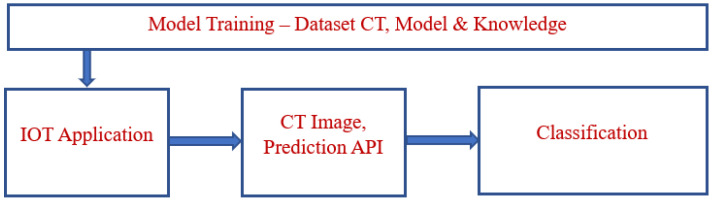
Classification of the system design model.

**Figure 6 bioengineering-10-00138-f006:**
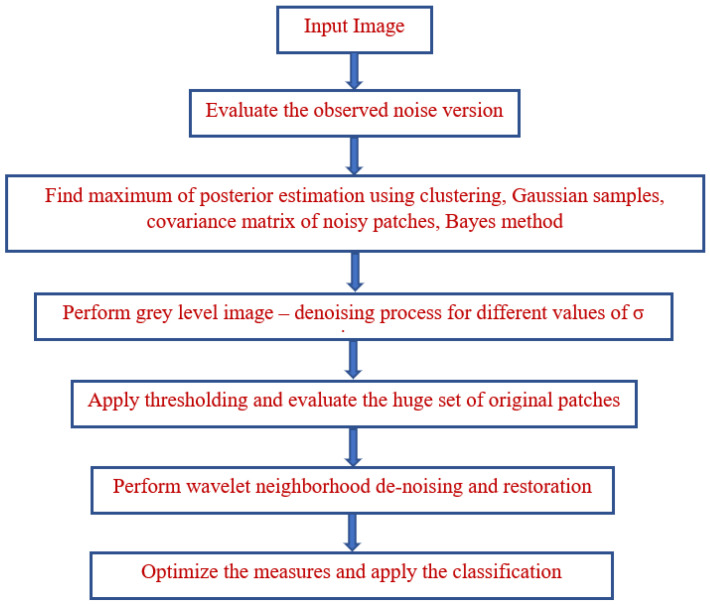
Flowchart of the proposed model.

**Figure 7 bioengineering-10-00138-f007:**
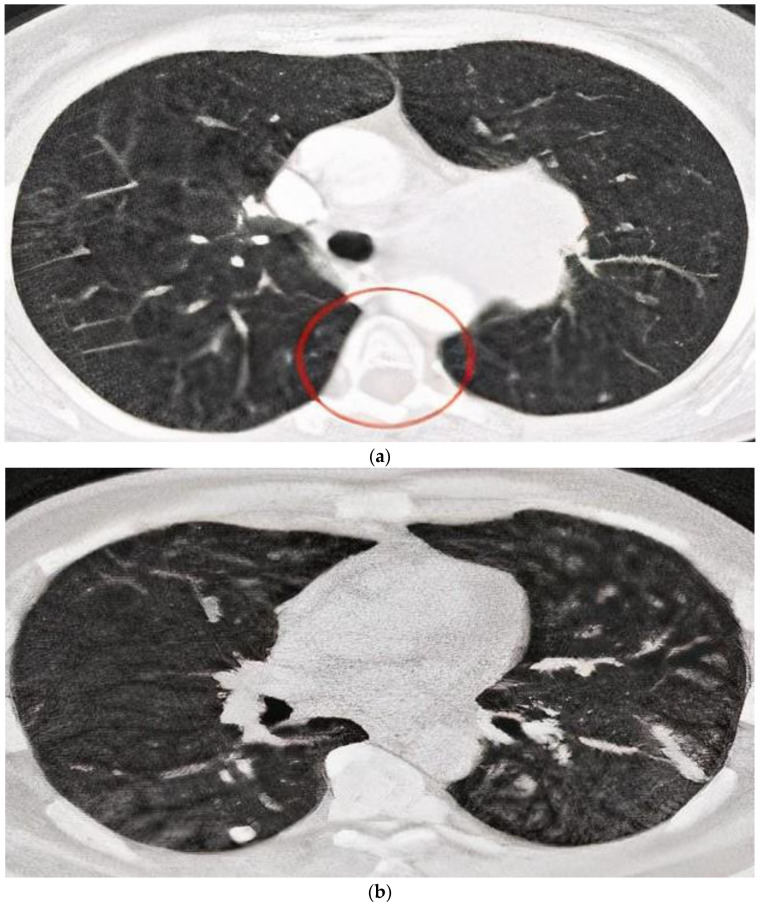
(**a**) CT scan image from the preprocessing stage. (**b**) CT scan image of noise removal.

**Figure 8 bioengineering-10-00138-f008:**
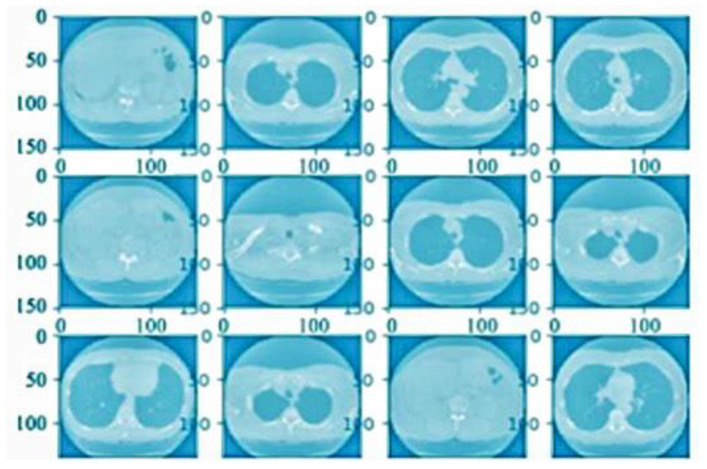
CT scan image after the normalization process.

**Figure 9 bioengineering-10-00138-f009:**
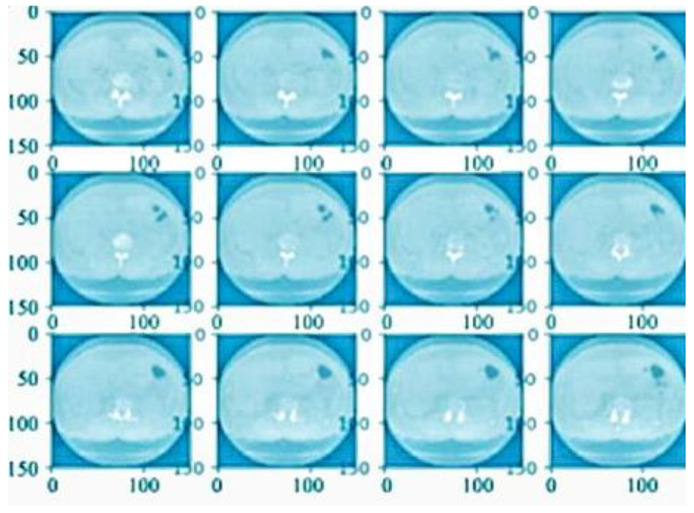
The finalized outcome of lung cancer prediction.

**Table 1 bioengineering-10-00138-t001:** Features of information types.

Types	Feature Name
Texture features	Correlation, Contrast, Homogeneity, Sum of square variance, Spectral, Spatial, and Entropy
Shape features	Area, Irregularity, Roundness, Perimeter, Circularity
Intensity features	Intensity, Mean, Standard Variance, Kurtosis, Skewness, Median
Geometric features	Eccentricity, Compactness, Roughness, Local Area Integral Invariant, Radial Distance Signatures

**Table 2 bioengineering-10-00138-t002:** Performance comparison with other models [1,2,3,4,5,6,7,8,41,42] for ChestX-ray14.

Parameters	Other Models	Proposed Model
Accuracy	84.02%	92.2%
Specificity	85.34%	98.8%
Sensitivity	82.71%	99.4%

**Table 3 bioengineering-10-00138-t003:** Performance comparison with other models [1,2,3,4,5,6,7,8,41,42] for JSRT.

Parameters	Other Models	Proposed Model
Accuracy	(58%, 72%)	(81%, 95%)
Specificity	(60%, 85%)	(83%, 95%)
Sensitivity	(24%, 67%)	(80%, 95%)

**Table 4 bioengineering-10-00138-t004:** Performance comparison with other models [1,2,3,4,5,6,7,8,41,42] for ChestX-ray14 and JSRT.

Parameters	Other Models	Proposed Model
Accuracy	(68%, 80%)	(82%, 89%)
Specificity	(65%, 83%)	(80%, 98%)
Sensitivity	(59%, 89%)	(85%, 98%)

## Data Availability

Not applicable.

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
