# Peer review of "Multi-Layered Non-Local Bayes Model for Lung Cancer Early Diagnosis Prediction with the Internet of Medical Things"

_bioengineering, 2023, doi:10.3390/bioengineering10020138_

Round 1

Reviewer 1 Report

In this manuscript, the authors aim to develop a framework implemented by Internet of Things (IoT) and Deep Learning (DL) to identify lung cancer disease, in which the proposed model used a DL process with a multi-layered Non-Local Bayes (NL Bayes) model to manage the process of early diagnosis. The expected accuracies of this proposed model are also analyzed and discussed. Though the methods and results are sequentially presented, the manuscript still needs many improvements:

(1) Line 6-Line 13: It is not necessary to list the institution corresponding to each author on by one, the authors with the same affiliation should be indexed the same superscript number.

(2) All of the figures in this manuscript are not clear and need to be improved (at least 300dpi).

(3) Section 3.4: The section needs to be polished and revised, especially for these equations, and a flowchart of proposed model needs to be added corresponding to Line 206-273.

(4) Line 206-273: the equations should be double checked and revised with professional formula editing software, such as word equation tools or Mathtype.

(5) Line 209: what is the meaning of “n” in Eq. (1)?

(6) Line 211-212: This is an extra left parenthesis in Eq. (2), what is the difference of the subscript “0”, “o” and “O” here and the other place? Is this a typo?

(7) Line 214: what is meaning of “C.” here?

(8) Line 215: the symbol of Euclidean norm should be used the professional expression.

(9) Line 216: Please pay attention to the distinction between regular and italic in the formula.

(10) Line 219: Please double check the expression of this equation.

(11) Line 221 and 223: the symbol of Euclidean norm needs to be revised and symmetrical.

(12) Line 236: what is the meaning of superscript “-1” in Eq. (11)?

(13) Line 241: This is an extra right parenthesis in Eq. (14).

(14) Line 247: what is the meaning of symbol “[]” here?

(15) Line 255: please double check the express of this equation, especially for the first “-1”.

(16) Line 257 and hereafter: what is meaning of “e.”? the form of Bayesian formula should be carefully checked.

(17) Line 300-301 and 309-310: The equation needs to be double checked and revised.

(18) Line 337: Some discussions should be added before the conclusion.

Author Response

Responses to Reviewer 1 is attached in the file

Reviewer 2 Report

1.) All the figures should be replaced with higher quality figures. The current provided figures are blurred and if these figures are taken from another source proper references should be provided.

2.) step 6 should be rewritten

3.) How did the authors define the range for patch size?

4.) For 2nd case of e1 the range should be 20<sigma<50 and same goes for the 2nd case of e2 the range should be 50<sigma<70

5.) Training and testing related information is completely missing from the manuscript. The information regarding the dataset should also be provided.

6.) An original figures for Fig. 6 and 7 should be provided for readers to understand the effect of applied image processing

7.) Results should be discussed further in detail.

8.) A discussion in FP and FN should also be included in the manuscript.

Author Response

Responses to Reviewer 2 is attached in the file

Reviewer 3 Report

This manuscript describes a computational comparison process in developing a framework to identify lung cancer disease. The authors concluded that the proposed model predicts images with high sensitivity and better precision value when compared with the specific result.

A few suggestions for the authors.

1.       Some of the references are not cited in the text with the right format. Examples like: page 2, line 60-61 “The method of the DL process to predict lung cancer with categorization and classification is discussed in [6]”; page 3, line 95, “of lung cancer with different types is discussed in [10]”; page 3, line 103, “The lung cancer detection process based on a DL system is discussed in [11]”.

2.       On page 4 and page 5, figure 1, figure 2 and figure 3 are not very clear. Similarly on page 7, figure 5 is also not very clear. Better images with higher resolution will be needed.

3.       On page 12, table 2, table 3 and table 4, the authors listed “Performance comparison with other models” but no information about the other models. Brief introduction about the other models will be needed so that the readers have some ideas about the improvements of the new model.

Author Response

Responses to Reviewer 3 is attached in the file

Round 2

Reviewer 1 Report

Since the authors have revised the manuscript by following my suggestions, I don't have other comments any more.